# Curcuminoids as Anticancer Drugs: Pleiotropic Effects, Potential for Metabolic Reprogramming and Prospects for the Future

**DOI:** 10.3390/pharmaceutics15061612

**Published:** 2023-05-29

**Authors:** Daniel L. Pouliquen, Koraljka Gall Trošelj, Ruby John Anto

**Affiliations:** 1Université d’Angers, Inserm, CNRS, Nantes Université, CRCI2NA, F-49000 Angers, France; 2Laboratory for Epigenomics, Division of Molecular Medicine, Ruđer Bošković Institute, 10000 Zagreb, Croatia; koraljka.gall.troselj@irb.hr; 3Molecular Bioassay Laboratory, Institute of Advanced Virology, Thiruvananthapuram 695317, India; rjanto@iav.res.in

**Keywords:** curcuminoids, curcumin, cancer, signaling pathways, metabolic reprogramming, chemosensitization

## Abstract

The number of published studies on curcuminoids in cancer research, including its lead molecule curcumin and synthetic analogs, has been increasing substantially during the past two decades. Insights on the diversity of inhibitory effects they have produced on a multitude of pathways involved in carcinogenesis and tumor progression have been provided. As this wealth of data was obtained in settings of various experimental and clinical data, this review first aimed at presenting a chronology of discoveries and an update on their complex in vivo effects. Secondly, there are many interesting questions linked to their pleiotropic effects. One of them, a growing research topic, relates to their ability to modulate metabolic reprogramming. This review will also cover the use of curcuminoids as chemosensitizing molecules that can be combined with several anticancer drugs to reverse the phenomenon of multidrug resistance. Finally, current investigations in these three complementary research fields raise several important questions that will be put among the prospects for the future research related to the importance of these molecules in cancer research.

## 1. Introduction

Turmeric (*Curcuma longa*) was used for thousands of years in traditional Indian and Eastern Asian medicine. Its cultivation in the Middle East was documented since the 18th century BC in the gardens of Babylon, well before its transfer to Africa, mainly through Arabic influences [1]. Finally spread to Europe thanks to Marco Polo, and then to America and Oceania via the international spice trade, it is one of the best examples of how the exchange of knowledge between different parts of the world benefited humankind in improving the health of millions of people. Among natural anticancer products, it also represents a fascinating and continuous link between past, present and future [1]. Its benefits as a food additive for the prevention of many diseases worldwide began to be documented through the report, in 1949, of the antibacterial action of its main component, curcumin [2]. However, a short report published in 1876 [3] demonstrated that a great portion of empirical knowledge accumulated over centuries in Asia was not fully valorized between these two dates. Nevertheless, its potential anticancer activity was originally demonstrated against the development of tumors in animals [4].The relief produced by a topical application of curcumin ointment in patients with external cancerous lesions was also revealed [5].

At the beginning of 2023, the number of publications related to the use of turmeric’s main phytochemical as an anticancer agent reached 7267 in the PubMed database. Although curcumin represents “the leading” molecule, a significant part of these articles refers to “curcuminoids” [6].The term also covers other plant secondary-metabolites belonging to the diarylheptanoids class [7], semi-synthetic structures such as prodrugs consisting of promoieties attached to the phenolic hydroxyl groups [8] and synthetic chemical molecules [9] that were initially produced for optimizing pharmacological potency [10]. Synthetic chemistry based on curcumin represents an exciting field of research, one that started with initial attempts: incorporating parts of the curcumin structure into an elaborate chemical scaffold, or trying to shorten or lengthen the aliphatic chain linking the two aromatic rings [11]. It has been developing since 2008 [11], and continues to grow [12].

Ten years ago, the diversity of inhibitory effects produced by these molecules on a multitude of molecular targets (e.g., enzymes, transcription factors and nucleic acids) and pathways involved in carcinogenesis and tumor progression started to be documented. The interest for curcuminoids in cancer research, with relation to diversity of targets and consequences, was explained by the different conformations they adopt to maximize these interactions, mainly through the keto-enol tautomer, flexible α, β-unsaturated β-diketo dimer, and the terminal o-methoxyphenolic groups [13].

This background event was the milestone for interesting prospects in the search for alternative and less toxic therapies against the most common malignancy worldwide and leading cause of cancer-related deaths, lung cancer [14]. However, at that date, most data were preclinical, including many in vitro studies. This weakness has been overcome during the last decade through insights provided by numerous in vivo and ex vivo investigations conducted on diverse experimental tumor models in laboratory rodents, or on clinical tumor tissue explants. Thus, this review will first aim at providing an update on this field. The potential of curcuminoids to reverse the deleterious consequences of the epithelial-to-mesenchymal transition (EMT) in invasive cancers was also recently reviewed [15]. However, among the many remaining unanswered questions regarding pleiotropic effects of curcuminoids, investigations focused on their suppressive effects on metabolic reprogramming, which were first documented ten years ago [16], represent a current growing research topic [17].

To avoid the consequences of curcumin’s hydrophobicity and low bioavailability, considerable efforts have been made in the production of a great diversity of drug delivery systems, which were recently extensively reviewed [18,19,20,21]. To reverse multidrug resistance, curcuminoids can be used in combination with many other drugs as chemosensitizer in cancer chemotherapy [22]. Thus, this crucial research field that continuously opens increasing potential will also be covered in this review. In this review, we have tried to profile available data very carefully, especially when interpreting data with respect to cell lines used, as there is an increasing problem with propagating and publishing results using the cell lines that are commonly contaminated with HeLa [23].

## 2. Evidence of Multi-Targeted Action of Curcuminoids against Cancers In Vivo: Historical Overview and Recent Findings

The main signaling pathways/pharmacological actions affected by curcuminoids in cancer are summarized in Table 1.

### 2.1. Historical Overview

Three successive research phases can be distinguished when studying the findings reported in in vivo models over the decades that followed the pioneering works of Kuttan et al. [4,5].

#### 2.1.1. Chemopreventive Properties

At the beginning of the 1990s, the role of curcumin as a potent inhibitor of arachidonic acid-induced inflammation in vivo in mouse skin was revealed, together with an associated suppression of chemical-induced tumor promotion [70]. In parallel, another research effort, one that included rats fed by curcumin and then exposed to two different carcinogens administered intraperitoneally (i.p.), demonstrated its strong antimutagenic action in vivo [71]. Subsequently, studies on chemical-induced forestomach and skin tumors in mice reported that the anticarcinogenic activity of different natural curcuminoids could be explained in part by an alteration of the activation of the carcinogen metabolism [72]. One important discovery was a significant decrease in the number of both chemical-induced mammary tumors and in vivo formation of adducts in rats that were receiving curcumin intraperitoneally [54]. The chemopreventive properties of curcumin given orally during the initiation and/or post-initiation stage against azoxymethane-induced colon tumors in rats were also correlated to an increased rate of apoptosis in tumor cells (i.e., shrinkage, nuclear condensation and presence of apoptotic bodies) [34]. The previously mentioned inhibition of the arachidonic acid metabolism was then attributed to a stimulatory effect of curcumin on the expression of genes encoding three stress proteins: Hsp27, Hsp70 and αB crystallin [73]. Finally, the action of orally administered curcumin during the promotion/progression stage of colon carcinogenesis in rats was demonstrated, and the modulation of apoptosis associated with the molecular action of curcumin was confirmed in colon tumors [35]. During that period, the apoptotic process in cancer cells, as induced by curcumin, was also reported to involve the activation of caspase-3 and the generation of reactive oxygen species (ROS) in a rat model of histiocytic tumor [41].

#### 2.1.2. Modulation of Immune Functions and Inhibition of Signaling Pathways

The second phase started with an important finding in the field of immuno-oncology. Using the AK-5 rat tumor model (histiocytoma), Bhaumik et al. revealed a differential activation status in host macrophages and NK cells induced by curcumin during the spontaneous regression of the tumor after subcutaneous injection of AK-5 cells [74]. This study represented a bridge within investigations focused on the pharmacology of these molecules, investigations that were previously conducted separately in the fields of inflammation/immunology and oncology. Other reports made a functional link between the modulation of lymphocyte-mediated immune functions produced by curcumin and the induction of curcumin-related antitumor response in a mouse strain that spontaneously developed multiple intestinal adenomas [36]. Another breakthrough was the demonstration of curcumin’s action in the protection of the host (tumor-bearing mice) from tumor-induced immunosuppression and toxicity [40]. 

The hypothesis that curcumin may suppress tumor promotion through inhibition of different signal transduction pathways was also proposed, while questions regarding the pharmacological activities of its major metabolites after biotransformation emerged [75]. In this context, one of the first reviews of the anti-carcinogenic activity of curcumin in animal models suggested that its action includes suppression of oncogenes expression and impairment of the ubiquitin–proteasome pathway [76]. Curcumin was reported to induce histones H3 and H4 hypoacetylation in glioblastoma cells, leading to hypoacetylation-enhanced apoptotic cell death. On the other hand, its application induced neurogenesis of neural progenitor cells (NPCs) through suppressing differentiation into astrocytes while promoting differentiation into neurons. These two neurogenic phenomena were confirmed in vivo [77]. Interestingly, these findings on the anti-carcinogenic effects of curcumin resonated with preliminary studies conducted 12 years earlier, in China [78]. In the same year, the inhibition of NF-κB activation and increased apoptotic rate was documented after i.p. administration of curcumin to nude mice bearing xenografts of human prostate cancer (PC-3 cells) [48]. This work was closely related to the previous research conducted by another research group showing the inhibition of angiogenesis associated with orally given curcumin in another type of prostate cancer (LNCap) grown in nude mice [49]. Evidence of a decreased expression of cyclin D1 and phospho-IκB-α after curcumin treatment of mice bearing head and neck squamous cell carcinoma was published four years later [24].Over the next 8 years, these discoveries were further confirmed in many other rodent and human tumor models, and the number of reports published annually increased exponentially. Meanwhile, the signaling pathways affected by curcuminoids were extended to TP53, MAPKs, Akt, Notch-1, Nrf2, JAK/STAT, Wnt/β-Catenin, AMPK/COX-2 and others [79]. From 2008, these in vitro data started to extend into the clinic through clinical trials [80], which have also been recently reviewed [17].

#### 2.1.3. Extension of Preclinical Studies and First Clinical Trials

This stage corresponded to an extension of investigations in the field of immuno-oncology, joined with the research focused on the diverse signaling pathways previously shown to be targeted by curcuminoids. During that period, the rationale of using a single cancer drug for a single target was also questioned [28]. Attempts to use curcumin combined with immunization, using soluble proteins isolated from viable tumor cells, led to an increase of the anti-tumor humoral immune response and inhibition of B16-R melanoma cells’ growth in mice [28]. Such research approaches led to the positive assessment of the potential of this phytochemical to enhance the cytotoxicity of CD8^+^ T cells against lymphoma cells in mice, and to increase the accumulation of transferred T cells in target tumors and caspase-3-mediated apoptosis [81]. Immunocompetent rats bearing a syngeneic peritoneal mesothelioma tumor model and treated with multiple injections of curcumin had residual tumors infiltrated with many CD8^+^ T cells and decreased intensity of immunohistochemical staining of IL-6 and vimentin [63]. Immunohistochemical analyses of xenografted tumor biopsies of curcumin-treated mice bearing a murine mesothelioma revealed apoptosis that was stimulated via an increase of the pro-apoptotic signals mediated by phosphorylated p38, MAPK and CARP-1 proteins [64]. 

In a model of human liposarcoma, important insights into curcumin-induced apoptosis included discovery of ER stress which preceded the apoptosis and was induced by interaction with the sarcoplasmic/endoplasmic reticulum Ca^2+^-ATPase 2 (SERCA2) [42]. The same process of activation of the ER stress-signaling pathway was further confirmed for a synthetic curcuminoid in a mouse model for human lung carcinoma [82]. Finally, the inhibition of SERCA2 activity by a second-generation synthetic curcuminoid was later confirmed in two different models of triple-negative breast cancer in mice. In triple negative breast cell tumor tissues, an induction of SERC2-related autophagy was also reported after application of a second-generation curcumin analog, RL71 [55].

Meanwhile, the understanding of the mechanisms of inhibition of other signaling pathways targeted by curcuminoids was improved. In ovarian carcinoma-bearing mice, curcumin treatment was shown to suppress pathways mediated by NF-κBand phosphorylated STAT3. These phenomena were associated with a decrease in VEGF, IL-8 and MMP-9 [52]. Curcumin-related reduction of the metastatic potential of prostate xenografts in mice is connected with inhibition of the feedback loop between NF-κB and the proinflammatory cytokines CXCL1/2 [50]. Neutrophil elastase was identified as a direct protein-target of curcumin in the process of lung cancer angiogenesis. Targeting neutrophil elastase was shown to have a negative impact on the signaling cascade HIF1α (hypoxia inducible factor 1 subunit alpha)/mTOR/VEGF/VEGFR [58].

During that period, curcumin was shown to inhibit the growth of a great diversity of human tumors in xenotransplant or ortho-transplant animal models. These studies were reviewed, together with the results of the first clinical trials [83]. Moreover, curcumin was shown to affect the growth of a rat experimental glioblastoma [67]. Head and neck squamous cell carcinomas represented one of the first cancer types in which curcumin application showed promising features relevant for the proposed therapeutic activity [84]. These effects were systematically reviewed [85]. Insights into curcumin’s effect on mammary cancer progression were provided through the demonstration of its synergistic action with β-interferon-induced upregulation of the anti-oncogenic protein GRIM-19 and inhibition of STAT3 transcription [53]. The increase of the reactive oxygen species and activation of FOXO3a, a member of the Forkhead Box Class O transcription factor, was reported as being closely related to the apoptosis induced by a curcumin synthetic analog T63 in a lung cancer model in mice [59]. With different curcumin analog, C-150, another report demonstrated that the modulation of the Notch/Akt signaling pathway was involved in curcumin-mediated effects in vivo [86]. An interesting observation was the enriched membrane localization of β-catenin, which correlated with the inhibition of tumor growth of a prostate cancer xenograft [51]. Subsequently, a confirmation of the suppression of the Wnt/β-catenin signaling pathway by curcumin was provided, showing a decreased expression of Wnt3a, LRP6, phospho-LRP6 and phospho-β-catenin, as well as C-myc, and surviving in gastric carcinoma xenografts [45]. Another independent study highlighted curcumin-related inhibition of both STAT3 andβ-catenin in gastric carcinogenesis [46]. The inhibition of STAT3, which was previously well documented in the chemoprevention of lung cancer [87], was investigated in more detail through hybridization of curcumin with a synthetic moiety to increase the ROS level. Using this hybrid, the treatment of nude mice bearing human MCF-7 breast cancer cells led to suppression of p-STAT3 and a decrease in Ki-67 immunostaining in the tumor tissues [56]. Regarding tp53, curcumin administration led to an increased expression and nuclear accumulation of tp53, and upregulation of p21 in mice bearing tp53 wild-type H460 xenografts, but not in those bearing tp53-null H1299. It was later shown that curcumin has an activating effect on the axis tp53-miR-192-5p/215-XIAP in lung cancer [60], which is relevant, and contributes to apoptosis. Another work complemented these findings by showing curcumin-induced apoptosis of head and neck squamous cell carcinoma in vivo through the ATM/Chk2/tp53-dependent pathway [25]. 

To end this section: three other major breakthroughs characterized this period. The first one was the discovery of curcumin’s ability to inhibit proteasome activity, a discovery which was originally obtained in colon cancer and confirmed later in other cancer types [37]. The second was a confirmation of the functional connection between the inhibition of the NF-κB pathway and epigenetic modulation induced by curcumin, through downregulation of DNA methyltransferase 1, which was discovered in a mouse model of acute myeloid leukemia [31]. The third finding was curcumin’s effect on ATP synthase [29], a crucial point related to the reprogramming of tumor metabolism, which will be developed in the second part of this review.

### 2.2. Recent Findings, and New Developments

Among recent findings, important improvements were reported at the interface of the immune system and cancer cells through research in immuno-oncology. Curcumin treatment contributed to the reinvigoration of defective T cells via multi-level immune checkpoint axis suppression in xenograft tumor models of head and neck cancer in nude mice [26]. Our understanding of the complexity of curcumin’s actions against this cancer type also benefited from ex vivo studies on tumor tissue explants from patients. These studies confirmed curcumin’s role in the inhibition of NF-κB nuclear translocation and its immune stimulatory effect in the tumor microenvironment [88]. Using the same methodological approach, another interesting study showed how curcumin changed immune cell composition and localization in colorectal cancer and in adenoma-patient-derived explants [89]. In a liver cancer xenograft model in mice, curcumin treatment impacted another population of immune cells involved in immunosuppression, myeloid-derived suppressor cells (MDSCs) and inhibited the TLR4/NF-κB signaling pathway and the expression of inflammatory factors, including IL-6, IL-1β, prostaglandin E2 and cyclooxygenase-2 (COX2) [43]. These findings were complemented by the demonstration that a curcumin analog prevented the growth of a melanoma through the reduction of the population of Foxp3^+^Tregs tumor-infiltrating lymphocytes [30]. At this immuno-oncology interface, other insights were provided through the combined use of a model of aggressive peritoneal mesothelioma in immunocompetent rats [63] and high-throughput proteomic analyses applied to formalin-fixed paraffin-embedded tissues. In a first step, an analysis of abundance changes in 1411 liver proteins identified a set of proteins associated with the curcumin-induced immune response which included the purine nucleoside phosphorylase (PNPH), the enzyme that is involved in T cell functions [65]. Secondly, comparison of the mesenteric lymph node and liver proteomes obtained from the same rats revealed a high level of similarity with respect to the abundance of PNPH [90]. Finally, proteome analyses of tumor tissues from curcumin-treated rats revealed specific changes measurable through the abundance of 22 proteins regulating the tumor microenvironment, including a continuous rise in caveolin-1, a protein regulating immune cell infiltration, T cell activation and dendritic cell maturation [66]. 

Improvements are apparent when one follows the diversity of signaling pathways targeted by these molecules. The inhibition of growth and invasion of a human monocytic leukemia in mice was joined with alteration of MAPK and MMP-signaling [32]. In a glioblastoma model, the inhibitory action of curcumin on tumorigenesis in vivo was characterized by an inhibition of the p-AKT/mTOR pathway and enhancement of the expression of the tumor suppressor protein tp53 [68]. The mechanism of inhibition of the Wnt signaling pathway induced by curcumin in vivo was reported to involve the downregulation of axin2, a negative feedback regulator of this pathway [38]. The knowledge on the role of non-coding RNAs has been also progressing. In this field, a first insight related to curcumin was provided through discovery of its influence on the axis that includes hsa_circ_0007580 (circ-PRKCA), miR-384 and integrin subunit beta 1 (ITGB1) in the non-small cell lung cancer (NSCLC) model. Curcumin decreases circ-PRKCA, a sponge for miR-384, resulting in miR-384 increase and ITGB1 decrease, which, all together, leads to a decrease of the biological aggressiveness of cancer cells [61]. A second report documented the inactivation of the JAK2/STAT3 pathway in a liver cancer model, in a similar fashion: a curcumin analog, GL63, contributes to a decreased expression of circZNF83, a sponge for miR-324-5p, whose target is cyclin-dependent kinase 16 (CDK16) [44].

Important findings were also revealed in four additional fields. In relation to mitochondria dysfunction, a first finding corresponded to the generation of ROS and subsequent YAP-mediated JNK activation produced by a curcumin analog (while low protein levels of YAP were observed in breast cancer tissues) [57]. The accumulation of ROS, which, together with intracellular iron, represents a hallmark of ferroptosis, was reported to be specific for the action of curcumin in lung cancer in vivo [62]. To explain the strong effect of both curcumin and inhibitors of the mitochondrial sodium–calcium–lithium exchanger (NCLX) on colon cancer in vivo, a common mitochondria calcium overload was suggested as leading to mitochondrial membrane depolarization [39]. A second axis, represented by the activation of ER stress produced by curcuminoids, received additional insights through the report that demonstrated an increase of the expression of ATF4 and CHOP [47]. Two enhanced signaling pathways, PERK-eIF2a and IRE1a-XBP1, were also identified in glioma xenografts in mice following radiotherapy-induced immunogenic cell death [69]. A third research field was represented by curcumin’s inhibitory effect on proteasome activity, which, at the molecular level, benefited from the demonstration of the involvement of the p300/miR-142-3p/PSMB5 axis [27]. In the fourth research field, which included transcription regulation, curcumin was found to produce a reduction of the expression of the histone methyltransferase EZH2 in a xenograft mouse model of myelodysplastic syndrome [33]. An activation of the Nrf2 protein and the expression of its target, Hmox-1, was also observed in vivo in mice after topical treatment [91]. In a work dedicated to the understanding of the mechanism of interleukin 17-A mediated acute lung injury in mice, a proteomic analysis revealed that the increased level of several mini chromosome maintenance proteins associated with increased level of IL-17A can be reversed by curcumin [92].

Although strongly cytotoxic for cancer cells originating from various types of cancer, curcumin is not cytotoxic for non-tumorigenic cell lines in vitro, although some exceptions exist (Table 2).

## 3. Curcumin and Chemosensitization through Metabolic Reprogramming

In 2013, Vishvakarma et al. showed a decreased response of cancer cells (DL—Dalton’s lymphoma) to cisplatin and methotrexate if they were exposed to a high concentration of glucose. This phenomenon was associated with the high production of lactate, acidification of the medium, and increased expression of one of glucose transporters, GLUT1. Although the authors did not perform mechanistic studies, they concluded that the increased rate of cancer cells’ glycolysis contributes to a decreased rate of therapy response [99]. In a recently published paper [100], availability of glucose (up to 30 mM) and overexpression of GLUT1 were shown to be crucial for development of chemoresistance against doxorubicin and methotrexate in liver-cancer-originating cells HepG2.

One of the two provisional “emerging hallmarks” introduced in 2011 [101]—“cellular energetics”—is now described as “reprogramming cellular metabolism” or, commonly, “metabolic reprogramming” [102]. The metabolic reprogramming is a well-known feature of the cancer cell, a feature which develops for securing numerous cancer-specific properties. Cancer cells must sustain the activity of its biosynthetic machinery to support its own replication capacity and high mitotic rate. Its ability for metabolic reprogramming depends highly on availability of two major growth-supporting substrates: glucose and glutamine. 

In non-transformed cells, the precise regulation of glucose intake is under the strict control of growth factors [103] that develop a molecular communication network which extends to PI3K/AKT/mTOR pathway and results in an increased expression of transporters for glucose and amino acids [104]. The activity of this pathway is sufficient to increase the size of resting cells (G0). Malignant cells do not have to rely on this physiological regulation. Instead, their specific oncogenotype allows them to adapt their metabolism with regard to available nutrients in order to sustain their biomass and proliferation capacity through an increased rate of glycolysis.

Glycolysis takes place in the cytoplasm through a cascade of several enzymatic reactions, producing pyruvate which can be fermented to lactate in the cytoplasm or further oxidized in a series of reactions dependent on mitochondrial respiration. Fermentation of glucose to lactate is less efficient in terms of energy production, generating only two molecules of ATP per molecule of glucose. If oxygenation is sufficient, non-transformed cells convert glucose-derived pyruvate into acetyl-CoA and direct it into the citric acid cycle. Complete oxidation of glucose by oxidative phosphorylation (OXPHOS) follows, generating 36 ATPs from one glucose molecule.

If there is not sufficient oxygen, lactate dehydrogenase A (LDH-A), the enzyme that is commonly overexpressed in malignant tumors [105] converts pyruvate into lactate. It has been known for a while that promoter of LDH-A has a binding site for HIF-1α [106] which, especially in hypoxic environment, positively regulates the activity of the *LDH-A* gene [107]. However, oxygen availability is not the sole decisive factor in shifting glucose-derived energy production from OXPHOS to aerobic glycolysis, as many non-transformed, rapidly proliferating cells, and cells with high anabolic activity are highly glycolytic [108].

### 3.1. Cancer Cells and the Warburg Effect

Malignant cells, even if there is a sufficient supply of oxygen, commonly avoid coupling glycolysis with OXPHOS (a phenomenon known as the Warburg effect), with a consequential increase of lactate production and low production of energy. This apparent paradox, early on, led to a proposition that cancer cells’ mitochondrial respiration is disrupted [109]. It is now known that malignant tumors retain functional mitochondria [110] and that mitochondrial respiration is required for their progression and metastasis [111,112,113]. Thus, apparently aerobic glycolysis and OXPHOS are not mutually exclusive, as the Warburg effect may occur also in non-transformed cells, and it is now recognized as a metabolic state in which the cell meets short-time-scale energy demands [114]. Recently, another facet of the Warburg effect’s regulation has been elucidated. Luengo et al. have demonstrated that, in a situation where the NAD+ requirement for oxidation reactions exceeds the demand for ATP, cells preferentially use glycolysis, since mitochondrial respiration cannot sufficiently regenerate NAD+, a cofactor needed for catabolism of reduced nutrients (e.g., sugars and lipids) and synthesis of oxidized macromolecules (e.g., nucleotides and amino acids). Mitochondrial regeneration of NAD+ is coupled with ATP synthesis. The rate of ATP hydrolysis, by supplying ADP as a substrate for ATP synthesis, gauges the rate of NAD+ regeneration in mitochondria and determines the extent of aerobic glycolysis vs. OXPHOS participation in glucose metabolism, regardless of oxygen availability [115]. A genetic setting of the malignant cell, in combination with a complex epigenome, directs an activity of genes involved in cellular metabolism that enables cancer cells to become metabolically reprogrammed: they produce different nutrients (metabolic flexibility) and process them in various ways (metabolic plasticity) [116].

Although traditionally considered as a “metabolic waste”, lactate—produced from pyruvate—is now recognized as a oncometabolite which is involved in development of resistance to radiotherapy/chemotherapy [117]. However, lactate is only one of many factors that contributes to the development and maintenance of cancer hallmarks, including chemoresistance. It is the whole specific oncometabolic make-up of the cancer cell that allows and favors a straightforward glycolytic conversion of glucose-derived pyruvate into quite complex branching points. This is possible because every step of glycolysis results in the occurrence of a specific intermediary product which a metabolically plastic cancer cell can, in dependence on its genomic constitution, use for its own highly demanding biosynthetic needs. The level of glycolytic energy production is very low, but this is compensated for by a consequential increased uptake of glucose. The activity of a specific branch, of which only two are presented on Figure 1, is determined through rate-limiting enzymes and the balanced equilibrium of respective intermediates [118].

Curcumin, through its pleiotropic mode of action, has potential for influencing the glycolytic process, and most, if not all, of the metabolic branching points. In this, no doubt, broad and complex part of cancer cell biology, we tried to connect data related to select processes involved in cancer cell metabolism, primarily glycolysis coupled with serine synthesis ***de novo*** and resistance to applied therapy, with the beneficial effects of curcumin application that are associated with specific metabolic features of cancer cells. 

### 3.2. Curcumin: Glycolysis and Lactate Production

Improvement of therapeutic efficacy of chemotherapeutics in the presence of curcumin in various in vitro models, as well as in experimental animals, has been documented and reviewed numerous times [119]. Recent reports indicate curcumin’s strong influence on the decrease of lactate production and/or the inability of cancer cells to excrete lactate. Once excreted, lactate acidifies the tumor microenvironment, and contributes to aggressive behavior of malignant tumors [117]. An in vitro model of liver cancer and T cell lymphoma (HepG2 and HUT78) was explored to show that curcumin, even in low concentrations (5 μM/24 h), significantly reduces lactate concentration in the medium and, through that mechanism, contributes to increased sensitivity of HepG2 cells to doxorubicin and methotrexate. External addition of 20 mM lactate reversed the phenotype toward chemoresistance [100], and was associated with an increased activity of genes coding for proteins whose oncogenic potential was shown in various cancer models (STAT3 and HIF-1α), the earlier mentioned LDHA, lactate receptor—HCAR1 (hydroxycarboxylic acid receptor 1/GPR81) and ABCB1 (ATP binding cassette subfamily B member 1; also known as MDR1 or P-glycoprotein). In 2017, Wagner et al. showed that lactate, acting through its receptor—HCAR1, strongly increases expression and activity of ABCB1 through a yet unknown molecular mechanism, resulting in PKC-dependent decreased doxorubicin sensitivity of the HeLa cervical cancer cell line [120]. In that scenario, chemoresistance related to increased production of lactate may be negatively influenced by curcumin, not only through curcumin’s negative regulation of HCAR1 [120], but also through a negative regulation of the GLUT1 transporter and a consequential decrease in the cellular availability of glucose [121]. 

Proteomic study performed after electric pulse application of 50 μM curcumin (EP_Cur) to triple negative breast cancer (TNBC) cell line MDA-MB-231 demonstrated decreased level of glycolytic enzymes and, at the same time, an increase of enzymes involved in OXPHOS, which is associated with dramatic decrease of cellular lactate [122]. This discovery clearly demonstrates the multilevel action of curcumin. In colon cancer cell lines HT-29 and HCT116, applications of high concentrations of curcumin (40 μM) for 24 h also resulted in significant decreases of lactate production (39.1% and 34.5%, respectively) [123].

As shown in Figure 1, the first step of glycolysis is catalyzed by hexokinase. There are four highly homologous hexokinase isoforms in mammalian cells. AKT potentiates hexokinase activity [124], resulting in phosphorylation of glucose molecules and their retention in the cell.

Hexokinase 2 (HEX2) was shown to be highly expressed in various malignant tumors [125]. At least in vitro, the level of its expression is significantly different in triple-negative breast cancer TNBC (model: MDA-MB-231; high expression) compared to estrogen-receptor-positive breast cancer cells (model: MCF-7; low expression). High expression of HEX2 in TNBC is under direct transcriptional control of overexpressed transcription factor SLUG. High expression of SLUG and HEX2 strongly associates with resistance of MDA-MB-231 to 4-hydroxytamoxifen (4-OHT). The resistance can be ameliorated through combined application of curcumin and 4-OHT, leading to apoptotic death [126]. In colon cancer cell lines, HT-29 and HCT116, application of 40 μM curcumin for 24 h resulted in a decreased rate of glycolysis, which is associated with significant decrease of HEX2, and its dissociation from the outer mitochondrial membrane [123].

There is strong evidence that, in cancer cells, curcumin fatally affects oxidative phosphorylation and contributes to energetic deficit by inhibiting ATP-synthase activity, which is associated with significant increase of ROS [29]. On that expectation, it was hypothesized that, although exposed to curcumin, transformed cells may be able to replenish lack of ATP through switching to a less-efficient energy producing metabolic pathway—glycolysis. When testing enzymatic activity of glycolytic enzymes and extracellular concentrations of lactate in four murine cancer cell lines (i.e., L1210—lymphocytic leukemia, 4T1—breast, B16—murine melanoma and CT26—colon), it was demonstrated that cells exposed to curcumin have a significant decrease of activity of hexokinase, lactate dehydrogenase, phosphofructokinase and pyruvate kinase (which may indicate the switch from active PKM1 to less active PKM2—which will be discussed later), in all but melanoma-originating B16. At least for phosphofructokinase and lactate dehydrogenase A/B, this may be a consequence of their decreased cellular level—as demonstrated in some other experimental models [127]. There was a clear distinction between neuroectoderm-originating melanoma B16 and the other three cell lines, as all measured glycolytic parameters become significantly increased only in the B16 cell line, probably reflecting its specific origin.

In two human cell lines originating from glioblastoma (U-87 MG) and neuroblastoma (SH-SY5Y), exposure to curcumin and its analogue, MS13, induced significant changes in the levels of various proteins, including a few involved in metabolic regulation, namely, glyceraldehyde-3-phosphate dehydrogenase (GAPDH; decrease in both MS13-treated cell lines) and phosphoglycerate kinase 1 (PGK1) in MS13-treated SH-SY5Y [128]. In human leukemia cell lines, K562 and LAMA84, exposure to curcumin was associated with severe changes in numerous proteins (143 upregulated and 234 downregulated). Among the decreased proteins were phosphoglycerate mutase 1, phosphoglycerate kinase 1, D-3-phosphoglycerate dehydrogenase (a rate-limiting enzyme of the serine de novo biosynthetic pathway) and pyruvate kinase (PKM) [129]. Curcumin can directly target metabolic enzymes; this was shown in another study, one based on cell-permeable clickable curcumin probe and quantitative chemical proteomics in a colon cancer cell line, HCT116 [130]. A stringent profiling revealed the following metabolic enzymes as curcumin binding partners: pyruvate kinase isozymes M1/M2, fructose-bisphosphate aldolase A, glyceraldehyde-3-phosphate dehydrogenase, alpha-enolase (enolase 1), L-lactate dehydrogenase A and B chains, phosphoglycerate kinase 1, D-3-phosphoglycerate dehydrogenase, and mitochondrial serine hydroxymethyltransferase (SHMT2).

It is very interesting that phosphoglycerate kinase 1, in addition to being presented as curcumin’s binding partner, turned to be listed as downregulated in all the proteomic studies we could find that were related to cancer and curcumin. As recently reviewed, a high expression of phosphoglycerate kinase 1 is positively associated with chemoresistance in all cancer models explored so far [131]. In 2000, Elson et al. showed that HIF-1α, PGK1, GLUT1 and VEGF occur in the earliest phase (early-stage hyperplasia) of multistage epidermal carcinogenesis [132]. Only four years later, Li et al. showed that PGK1 has the HIF-1α binding site in its promoter and that transcriptional activity of PGK1, under hypoxic condition, depends on HIF1α [133]. This transcription factor has been associated with chemoresistance in various types of cancer [134]. Extracellular ATP can stimulate hypoxia-inducible factor (HIF) signaling and contribute to breast cancer cell resistance even under normoxic conditions [135]. The most recent data show that STAT-3—mediated ALDOA (fructose-bisphosphate aldolase A), binding to HIF1α (which itself contributes to multiple resistance in cancer) strongly contributes to development of chemoresistance against cisplatin (models: breast cancer cell line MCF-7 and xenografts MDA-MB-231 [136].

In these scenarios, curcumin indeed may—due to its pleiotropism—add to chemosensitization through synchronous actions at several molecular levels. For example: (a) through downregulation of ALDOA [122], (b) through negative influence on HIF1α, as recently reviewed [137], (c) through inhibitory effect on STAT3 (direct binding of curcumin to STAT3 was discovered a few years ago [138]). All these data clearly show that metabolic reprogramming of cancer cells, when in favor of glycolysis and lactate production, strongly increases chemoresistance of tumor cells. They also show a very strong, functional interconnection between the metabolic status of the cell and various signaling pathways that are not traditionally considered “metabolic”. This shows that none of the hallmarks of cancer should be considered as an isolated entity because molecules and processes related to cancer hallmarks are shared among various signaling pathways, and one pleiotropic molecule may indeed affect several of them at the same time.

### 3.3. Curcumin and Lactate Excretion

Glycolytically produced lactate must be excreted from the cell, where it contributes to extra-tumoral acidosis. It is known that malignant tumors maintain their hallmarks through maintenance of intracellular alkalinity (pHi ≥ 7.4) and extracellular acidity (pHe~6.7–7.1) [139]. These values are significantly different from values related to differentiated cells (pHi~7.2; pHe~7.4). This cancer-related phenomenon, known as “pH gradient reversal” was recognized in 1996 as a possible vulnerability that may be triggered by therapy [140], and was recently reviewed [141]. Lactate shuttle, which is needed for establishment of a synergistic metabolism between glycolytic tumor cells and tumor cells relying on OXPHOS, is highly dependent on the activity of lactate transporters MCT1 and MCT4 (SLC16A1 solute carrier family 16, members 1 and 4). In order to explore the effect of chemosensitization of the whole extract of the *C. longa*, and then each active compound separately: curcumin, bis-curcumin and demethoxycurcumin, with respect to cancer cell response to 5-FU and lactate metabolism, Li et al. [142] used the 5-FU resistant colon cancer cell line, HCT8, previously shown to overexpress MCT1 [143]. The authors were able to show that the chemosensitizing effect of the whole extract, in this experimental model, partially relates to a significantly decreased expression of MCT1 and high increase of intracellular lactate (three- and four-fold change), after application of curcumin and bis-curcumin, respectively. It is expected that so significant increase of intracellular lactate changes the intracellular pH (pHi), creating a scenario which may have a potential for being detrimental for cancer cells (toxic acidosis). In another model, application of curcumin was shown to have a potential for shifting the intracellular pH of cancer cells toward acidity through decreasing both MCT1 and Na+/H+ antiporter NHE1 (SLC9A1—solute carrier family 9 member A1 [144], which also may contribute to curcumin’s chemosensitizing properties.

As recently shown, increased intracellular acidity induced by hypoxia plus MCT1/2 inhibition significantly compromises the survival of MCF7 breast cancer cells. The effect was amplified when glycolytic enzyme, glyceraldehyde-3-phosphate dehydrogenase (GAPDH), was silenced [145]. Whether exposure to curcumin produces exactly the same mode of action remains to be explored. Obviously, pleiotropic action of curcumin can target cancer cell metabolism at many levels that are closely related and interconnected.

### 3.4. Curcumin and Pentose Phosphate Pathway (PPP)

Although not providing adenosine 5′-triphosphate (ATP) for cellular energy demands, high activity of PPP (Figure 1, in blue circle) in cancer cells makes a strong contribution to a reducing cellular power (through production of NADPH) and ribonucleotides synthesis (through production of ribulose 5-phosphate (R5P) [146]. Production of NADPH through PPP is needed for generation of reduced glutathione (GSH) which allows for alleviation of oxidative stress through successful removal of ROS [147]. This oxidative branch of PPP relates to the activity of two enzymes, glucose-6-phosphate dehydrogenase (G6P) and phosphogluconate dehydrogenase (PGD).

As recently reviewed, the increased activity of PPP relates to cisplatin chemoresistance that can be developed through several mechanisms [148]. The NADPH generated in the PPP counteracts, for example, a high Cisplatinum cytotoxicity associated with a high production of ROS. As might have been expected, inhibition of the two enzymes related to NADPH generation, G6P and PGD, makes cancer cells sensitive to cisplatin [149,150].

There are no data related to the selective effects of curcumin on G6PD specific to cancer cells. However, feeding animals with curcumin (2%, *w*/*v*) increases activity of G6PD in mice liver and kidney tissues for 89% and 67%, respectively, contributing to the systemic defense against oxidative stress, as expected [151], in a chemopreventive, and not a therapeutic, setting.

### 3.5. Curcumin, Pyruvate Kinase and Serine Synthesis Pathway (SSP)

In cancer cells, biosynthesis of serine (SSP—Serine Biosynthesis Pathway) depends on several interconnected factors: (a) availability of the precursor, 3-phosphogylcerate (glycerate 3-phosphate; labeled green on Figure 1, green circle) whose generation is dependent on glucose intake and presence of metabolically inert pyruvate kinase 2 (PKM2), (b) ATF4-directed expression of three metabolic enzymes, (1) PHGDH—Phosphoglycerate-3-Dehydrogenase; (2) PSAT1-Phosphoserine Aminotransferase 1; and (3) PSPH-Phosphoserine Phosphatase. 

There are numerous studies demonstrating an increase of each of these enzymes in various malignant tumors. There is also a study showing the increased protein level of all three proteins (fold change: PHGDH: +3.62; PSAT 1: +6.94: PSPH: +1.27) in lung cancer tissue, when compared to corresponding non-tumorous tissue [152]. In cancer cells, serine has an important role associated with the activity of pyruvate kinase (PKM).

Pyruvate kinase catalyzes the last glycolytic step: transfer of phosphate group from phosphoenolpyruvate (PEP) to ADP for generation of ATP and pyruvate. Contrary to isoenzyme PKM1, which is highly expressed in tissues with high energetic needs (e.g., heart, muscle and brain), expression of PKM2 is common in malignant tumors. It is considered a cancer-specific isoenzyme. The difference between PKM1 and PKM2 depends on alternative splicing of the primary transcript of the PKM gene.

The PKM2 activity depends on a highly complex, serine-dependent allosteric regulation: contrary to PKM2 tetramers that are strong catalyzers (as are PKM1 tetramers), PKM2 specific dimers are catalytically inert, and they support biosynthetic processes [153]. Lack of serine favors accumulation of inert PKM2 dimers, decreased generation of pyruvate and accumulation of glycolytic intermediary products of which one—3PG—enters the SSP (Figure 1). De novo synthesized serine may be converted to glycine through the activity of mitochondrialserine hydroxymethyltransferase 2 (SHMT2). Glycine is then used for synthesis of glutathione (GSH) and purine nucleotides. 

Thus, in its essence, the active SSP is beneficial for cancer cells, not only with respect to serine supply (when needed), but also for development of resistance to therapy, due to its positive impact on GSH synthesis, which is needed for cellular defensive response against therapy-induced oxidative stress. Curcumin binds to SHMT2 [154], and there is a high probability that the binding impacts both the structure and the catalytic activity of the active form of SHMT2 [155], which was recently shown to drive resistance to 5-FU in a colorectal model of cancer [156].

Increased level/activity of enzymes directly and indirectly involved in SSP and pyruvate kinase activity have been shown in various experimental models, and in native tumors. The importance of some of them for development of resistance to therapy was also shown in various cancer models. However, the available data is not necessarily unequivocal and asks for a careful interpretation with respect to the specifics of the cancer’s origin: high expression of PKM2 was shown to be associated with resistance to cisplatin in bladder cancer [157], but, on the other hand, it enhances response to cisplatin in cervical cancer through a complex interaction with the mTOR signaling pathway [158]. Although an indicator of worse clinical prognosis in breast cancer patients, high expression of PKM2 was shown to strongly associate with a positive therapy response to epirubicin and 5-fluorouracil, resulting in longer disease-free survival and overall survival [159]. Most of the studies related to the role of PKM2 in the setting of cancer therapy were recently reviewed [160].

The published studies demonstrate that curcumin’s effects on PKM2 are not entirely conclusive: Sidiqqui et al. [161] demonstrated that curcumin decreases the PKM2 level at the level of mRNA and the protein in cell lines of different origin (H1299—non-small cell lung cancer; MCF7—breast cancer; HeLa—cervical cancer). There are data showing that the type of response may be cell-type specific: for example, curcumin was shown to decrease PKM2 mRNA level in only one (Cal27), among three tested cell lines originating from head and neck carcinoma (Cal 27, FaDu and Detroit 562) [162]. Yadav et al. have shown that curcumin influences splicing of the PKM transcript in favor of PKM1 and, through that mechanism, reduces the level of PKM2 transcript and the corresponding level of inert PKM dimers [163].

### 3.6. Curcumin and Three Enzymes of the SSP

The limiting enzyme for serine de novo synthesis is NAD+-consuming PHGDH. Its high expression in cancer was originally shown to be associated with amplification of the chromosomal locus 1p12, where the gene resides, in melanomas and breast cancer [164]. During the last ten years, serine was recognized as a very important molecule involved in cancer cell metabolism, and the rate-limiting enzyme for its synthesis, the PHGDH, came in the focus of research as a potential therapeutic target. Many various PHGDH inhibitors have been developed and tested, but are still not making their way into the clinic [165].

In tissues with extremely low serine availability (brain tissue), inhibition of PHGDH was shown to be a very beneficial therapeutic approach: TNBC originating metastases cannot develop in brains of animals with inhibited activity of PHGDH [166]. A high level of PHGDH relates to resistance to cisplatin, 5-FU and Sorafenib in ovarian cancer [167], colorectal cancer [168] and liver cancer [169], respectively. Resistance to other types of cancer therapy, mediated through PHGDH, was reviewed recently [170]. Although commonly presented in a simple way, the metabolic scenario in which PHGDH exerts its pro-tumorigenic activity is multilevel and highly complex.

As recently demonstrated [171], a strong association between relapse of disease in a cohort of ovarian cancer patients receiving Cisplatinum and, at the same time, having a low level of PHGDH, was well-documented. The phenomenon was explored on a molecular level: when exposed to cisplatin, cancer cells balance available NAD+ between PARP-1 (PARP-1 activity is needed for DNA repair) and PHGDH in favor of the repairing mechanism mediated by PARP1 and, instead of activating SSP, take serine from the medium. Indeed, in experimental animals, a diet with decreased content of serine/glycine combined with selective inhibition of PHGDH was shown to have a very potent effect with respect to inhibition of cancer growth, in vitro, and in vivo [172].

PHGDH has been shown to be the curcumin binding partner in three experimental systems: benign schwannoma [173], HeLa (cervical cancer) [154], and HCT116 colon cancer cells [130]. One would expect that curcumin bound to PHGDH enzyme NAD+ pocket abolishes its activity and negatively influences pro-proliferative SSP. Although there are many experimental data which show an association of PSAT1 and biological behavior of malignant tumors, there are sparse data on its involvement in chemoresistance. That is surprising because the activity of this second enzyme in serine de novo biosynthesis contributes to the development of chemoresistance to FOLFIRI (leucovorin, 5-FU and irinotecan) treatment. This was discovered in a small cohort of colon cancer patients as early as 2008 [174]. The most recent data indicates its association with the tumor immune microenvironment [175]. In glioblastoma multiforme, regorafenib (inhibitor of multiple tyrosine kinase) exerts its effect through stabilization of PSAT1, leading the malignant cell into autophagy. In this scenario, regorafenib may induce autophagy only if a high level of PSAT1 exists in the cell [176]. 

The third SSP enzyme, phosphoserine phosphatase, is associated with poor therapeutic response of ER+ breast cancers to tamoxifen [177], and acts as an oncogene in various types of malignant tumors, including non-small-cell lung carcinoma (NSCLC) [178].

In 2015, Chiang et al. showed that exposure to curcumin reduced the gene activity of all three SSP enzymes (PHGDH (−3.19), PSAT1 (−2.27) and PSPH (−2.20)), using the model of lung cancer cell line, NCI-H460 [179].

## 4. Curcumin as a Chemosensitizer in Conventional Chemotherapy

### 4.1. Chemosensitization: Need of the Hour

Natural products from diverse sources encompass a unique arena of chemical space that overlaps extensively with pharmaceuticals which are not found in synthetic chemical libraries. These natural compounds have been evaluated for a long time for their ability to selectively target different kinds of ailments, including cancer. Most of the anticancer drugs used in chemotherapy are natural-product-derived compounds [180,181]. Nowadays, most of the currently available chemotherapeutics fail to accomplish their expected outcome. Upon prolonged exposure to the chemotherapeutics, the emergence of inherited and acquired chemoresistance due to the upregulation of major survival signals is regarded as the main cause for this drawback. Combination chemotherapy is currently used for various cancer types, and this approach is more effective, compared to single-agent treatment [182]. Multidrug resistance (MDR), the ability of tumor cells to develop resistance to a broad range of structurally and functionally unrelated drugs, is also a major hindrance in the success of combination chemotherapy. Apart from the classic mechanisms of MDR development (over-expression of drug efflux pumps and increased activity of DNA repair machinery), alterations at the level of apoptosis control serve as a crucial mechanism for the induction of drug resistance [180]. Increasing the dosage of the drug can certainly help in evading the condition, but serious side effects limit the success of the clinical outcomes of chemotherapy. It may also lower patients’ quality of life, and, in some cases, even result in discontinuation of chemotherapy [183].

Thus, chemosensitization can be a choice that absolutely matches the need. Chemosensitization can be defined as a process by which a non-toxic compound of either natural or synthetic origin sensitizes the cancer cells to a cytotoxic therapeutic agent without affecting the efficacy of the same. This has an added advantage of minimizing the dosage of the chemotherapeutic and thereby decreasing the side effects. Additionally, this strategy is more economical, considering the cost of currently available chemotherapeutic drugs, especially in the case of developing countries, where the rate of cancer incidence is relatively high. Since chemoresistance is a tightly regulated process under the control of multiple survival pathways, the inhibition of any single molecule may not be sufficient to circumvent the phenomenon. Hence, compounds that can simultaneously modulate multiple survival-signaling pathways might provide a better therapeutic outcome than that of individual inhibitors. Several phytochemicals have been shown to modulate multiple pathways involved in chemoresistance and, hence, are assumed to be of better chemosensitizing efficacy.

### 4.2. Curcumin: The Celebrity among Nutraceuticals

From ancient days onwards, Indian and Chinese traditional medicine have made use of combinations of medicinal herbs. Among those, flavonoids are a large subgroup of the family of natural polyphenolic compounds, which are mostly the part of secondary metabolism in plants [184,185]. Most of the plant polyphenols, including curcumin, resveratrol, genistein, quercetin, epigallocatechin gallate (EGCG), luteolin, apigenin, chrysin, tannic acid, etc. are dietary compounds which are part of our day today diet. Amongst the wide range of natural polyphenols, curcumin is the most-studied natural compound, with perfect documentation of its therapeutic effect in a large number of disease conditions, including cancer [186,187].

Curcumin is well known for its chemopreventive [84,188,189,190,191,192,193,194], as well as chemosensitizing [22,180,195,196,197,198,199,200,201,202], efficacy against cancer, together with anti-inflammatory, antioxidant and antibacterial activities [203,204]. We have attempted to compile the studies in which curcumin was used as a chemosensitizer [180]. 

### 4.3. Molecular Targets of Curcumin as a Chemosensitizer

Curcumin exerts its anticancer effect through targeting various regulatory molecules, including protein kinases, transcription factors, receptors, enzymes, growth factors, cell cycle, and apoptosis-related molecules, as well as microRNAs. Several reports have shown the effects of curcumin on variety of key molecular signaling pathways, such as NF-ĸB, MAPK, PI3K/Akt/mTOR, JAK/STAT, Wnt/β-catenin, etc. [205,206]. It also possesses modulatory effects on the apoptotic, metastatic and cell cycle pathways involved in cancer development and progression [196]. The major molecular targets that regulate the chemosensitizing efficacy of curcumin are depicted in Figure 2.

### 4.4. Curcumin as a Chemosensitizer in Adjuvant Chemotherapy

Several in vitro, in vivo and clinical trials have shown the chemosensitizing efficacy of curcumin in combination with current chemotherapeutic drugs. Curcumin has been reported to potentiate the antitumor effects gemcitabine via downregulation of COX-2 and phospho-extracellular signal-regulated kinase1/2 (ERK1/2) levels in pancreatic adenocarcinoma cells [207], as well as through the inhibition of gemcitabine-induced NF-κB and its downstream targets, in an orthotopic model of pancreatic cancer [208]. A phase II clinical trial also evaluated the effectiveness of the combination [209]. Studies have shown that curcumin sensitizes breast cancer cells to 5-FU-mediated chemotherapy through the inhibition of 5-FU-induced upregulation of thymidylate synthase (TS), both in vitro [210] and in vivo [197], irrespective of the receptor status. It was also shown that the antitumor effects of paclitaxel could be enhanced by curcumin in cervical cancer cells through the downregulation of paclitaxel-induced activation of NF-κB, Akt and Bcl-2 [22,201,202]. The ability of curcumin to augment the antitumor effect of capecitabine in human colorectal cancer by modulating cyclin D1, COX-2, matrix metallopeptidase 9 (MMP-9), VEGF and C-X-C chemokine receptor type 4 (CXCR4), has been assessed by using an orthotopic mouse model [211]. Curcumin has also been also shown to sensitize prostate cancer cells to the cytotoxic effect of 5-FU through a tp53-independent cell-cycle arrest and the downregulation of constitutive NF-κB activation [212]. Curcumin enhances the cytotoxic effects of 5-FU and oxaliplatin in colon cancer cells through the downregulation of COX-2 and the modulation of EGFR and insulin-like growth factor 1 receptor (IGF-1R) [213]. Both in vitro [214] and in vivo [215] data shows that curcumin-mediated inhibition of NF-κB activation enhances the sensitivity of prostate cancer cells to TRAIL-induced apoptosis. It has been reported that the combination of curcumin and doxorubicin could enhance the sensitivity of breast cancer cells through inhibition of ABC subfamily B member 4 (ABCB4) activity [216]. Downregulation of IAPs by curcumin has also been reported to enhance the effect of cisplatin in hepatic cancer cells [217]. Recent reports have also suggested that the combination of curcumin and paclitaxel could inhibit the ALDH-1 and paclitaxel-induced Pgp-1 expression in breast cancer cells. The combination resulted, in treated cells, in upregulation of Bax, caspase-7, and caspase-9, along with downregulation of Bcl-2 expression [218]. Curcumin has been shown to inhibit the FA/ BRCA pathway, and it sensitizes ovarian cancer cells to cisplatin-induced apoptosis [219]. It has shown that inhibition of HER2 and reduction of NF-κB activation by the combination of curcumin and its derivatives with doxorubicin enhances the toxicity of doxorubicin in resistant breast cancer cells [220]. Moreover, doxorubicin-induced over-expression of major proteins, including vimentin, β-catenin, p-AKT, p-Smad2 and p-GSK3β, Snail and Twist, which are involved in EMT and metastases of TNBC cells, were found to be downregulated through the suppression of TGF-β and PI3K/AKT signaling pathways [221]. Studies have illustrated that co-treatment with curcumin and cisplatin sensitizes breast cancer cells to cisplatin through the activation of the autophagy pathway. In treated breast cancer cells, the key mechanism underlying the curcumin-mediated chemosensitization was found to be the downregulation of CCAT1 expression and inactivation of the PI3K/Akt/mTOR pathway [222]. Several reports have indicated the ability of curcumin to reverse the cisplatin resistance in lung cancer. It has been shown that curcumin enhances sensitivity of human NSCLC cell lines toward cisplatin treatment through influencing a Cu-Sp1-CTR1 regulatory loop [223]. Curcumin reverses cisplatin resistance and promotes human lung adenocarcinoma A549/DDP cell apoptosis through HIF-1α and caspase-3 mechanisms [224]. Results from a very recent study also showed that curcumin sensitizes NSCLC cells to cisplatin-mediated cell death through activation of the ER stress pathway [225]. Tan and Norhaizan showed that a combination of camptothecin and curcumin-loaded cationic polymeric nanoparticle increased intracellular drug concentrations and synergistic effects of the drugs in colon-26 cells [226]. Curcumin has been shown to prevent liver cancer stem cells’ growth through inhibition of the PI3K/AKT/mTOR signaling pathway [227]. Furthermore, curcumin-based nanoparticles and curcumin-tagged antibodies were reported as promising therapeutic strategies to overcome resistance in brain tumors [228].

## 5. Discussion

The three sections included in this article, which reviewed the main recent findings reported in the literature in some major areas of this research topic, raised several questions and prospects that need to be discussed. 

A first question related to the multiple signaling pathways and components of the tumor microenvironment targeted by curcuminoids is: what kind of cognitive tools would be required for understanding its complex, pleiotropic mode of action against the hallmarks of cancer? Although a wealth of data has been accumulated, both in vitro and in vivo during recent decades, many points are still lacking in our vision of how the complex regulatory networks governing the tumor microenvironment’s functions are modulated. To give just one example, the recent discovery showed the way curcuminoids act against the deleterious effects of cancer associated fibroblasts (CAFs) in invasive cancers [15]. New developments continue to add more complexity to our understanding of the diverse functions of this TME component [229]. CAFs increase matrix stiffness, promoting EMT, and, as a consequence the physics of the interactions at biological interfaces is likely to be changed under their influence. Cell–cell interfaces include transient interactions between mobile immune cells, but also long-lived cell–cell contacts, in which matrix stiffness may play a crucial role. However, the role of water, which indeed has been established as an active matrix, has never been considered in this scheme [230]. At the subcellular level, this important parameter is underrepresented in the molecular biology of the cell, despite major breakthroughs being published in the field [231]. Fortunately, recent developments and the emergence of innovative tools applied to cancer research [232,233] could contribute to answering this fundamental question in the future. Another related observation relates to the fact that curcuminoids, through their simultaneous action on multiple molecular targets, belong to the field of polypharmacology [234]. Thus, understanding their mode of action requires a more sophisticated view than the consideration of a single signaling pathway. Many questions come with this challenge, and there is a great potential for answering them through applying new approaches [235,236,237], especially in the field of newly emerged cancer hallmarks.

Hypoglycemic effects of curcumin in vitro have been well recognized. Promising antidiabetic effects were also shown in several clinical trials [238]. There are new data showing that curcumin, especially in glucose deprived cancer cells, decreases intracellular pH. The phenomenon was explored in a hepatocellular cancer cell line experimental model where curcumin showed an effect equally strong as the NHE1 inhibitor, cariporide [144]. Curcumin also decreased expression of lactate transporters MTC1 and MTC4. Thus, reversal of the pH gradient across the cell membrane, especially in the starved cells, may be the fundamental mechanism involved in curcumin’s action. These data are probably crucial for understanding the molecular background of various anticancer effects which have already been shown in different cancer cell lines exposed to a combination of curcumin and an antidiabetic drug, metformin [239,240,241]. A second question is: What resource would be best suited for the experimental study of curcuminoids’ effects against cancers? To deal with the limitations of the cancer-cell-lines’ monolayers, advances in in vitro 3D culture technologies are being continuously reported. However, even the success of organoids in cancer research [242] still raises questions related to the mode of investigation that should be applied for exploring the host’s immune response, as modulated by curcuminoid treatment in vivo There is a high level of complexity in the ecosystems represented by neoplastic tissues, comprising a heterogeneous population of tumor cells and a multitude of immune and non-immune cells communicating through a plethora of systemic mediators, and embedded in an extracellular matrix [243]. The metabolic status of various types of cells that already communicate in early tumorigenesis adds additional level of complexity. A potent oncometabolite, lactate, shuttles between cancer cells and stromal cells (reverse Warburg effect), hypoxic cancer cells and oxidative cancer cells (metabolic symbiosis), as well as between hypoxic cancer cells and vascular endothelial cells. The final outcome is occurrence of the fatal cancer hallmarks: angiogenesis, immune escape, cell migration, metastasis and self-sufficient metabolism [144,244]. Thus, the modelling represents an almost impossible task. This situation is even more complicated, given the fact that most stromal cells are exposed to dynamic changes during malignant progression [245].

Another question is related to an increasing number of reports demonstrating the role of interactions between curcumin given orally and gut microbiota, in the generation of active metabolites. This observation led to the “low bioavailability/high bioactivity paradox” that concerns not only curcuminoids but most dietary polyphenols [246]. In this new research field, enzymatic modifications of curcumin by bacteria have already been established and reviewed [247], and the consequences of the regulation of intestinal microbiota by curcumin with respect to chemotherapeutic treatment have begun to be investigated [248]. The question for the future of this research is: how to take advantage of this process for optimizing current therapeutic strategies when combination treatments are used?

An increasing number of reports on these molecules are published each year. They open many interesting prospects, which could be summarized in three main points. First, new active natural molecules belonging to this family are continuously being discovered in the world. To mention just a few, in the search for new marine anticancer drugs, and in a context of global climate change that could affect bioresources [249], the potential of new molecules extracted from marine invertebrates that have an ability to reverse the immune-escape phenotype in metastatic tumors has just started [250]. With respect to the third question discussed above, new biologically active derivatives produced through microbial transformation of curcuminoids (diarylheptanoids) isolated from natural resources are also being studied [251]. 

In parallel, new synthetic curcumin analogs are still being produced, some of them exhibiting innovative therapeutic potential against cancers. In this field, improvements are emerging in the identification of curcumin derivatives with improved binding affinities for some growth factor receptors [252]. Using another methodological approach, Pandya et al., studied the exact binding pattern that one analog of interest formed with the *c-myc* DNA sequence [253]. The way curcumin analogs boost the efficacy of anti-immune checkpoint inhibitors also represents a new research field under development [30]. Finally, among the 25 recently reviewed curcumin analogs [15], at least eight have been the subject of continuous investigations with respect to their pleiotropic anticancer effects, in 2023 alone [254,255,256,257,258,259,260,261]. As these molecules may overcome limitations of bioavailability and pharmacokinetics that were recently reviewed based on clinical data [17], the hope is that the current research might soon lead to clinical trials involving cancer patients for at least some of them.

Whether curcumin or curcumin analogs are selected for investigations of their impact on the different populations of immune cells involved in immunosuppressive microenvironment [262], there is a growing interest for the design and use of innovative nanoformulations [263]. Finally, among phytocompounds used to design and develop new drugs, curcuminoids are also being increasingly used as models with a potential for improving our understanding of their multiple modes of actions, using computational approaches such as virtual screening, target prediction, and molecular dynamic and pharmacophore modelling [264].

## Figures and Tables

**Figure 1 pharmaceutics-15-01612-f001:**
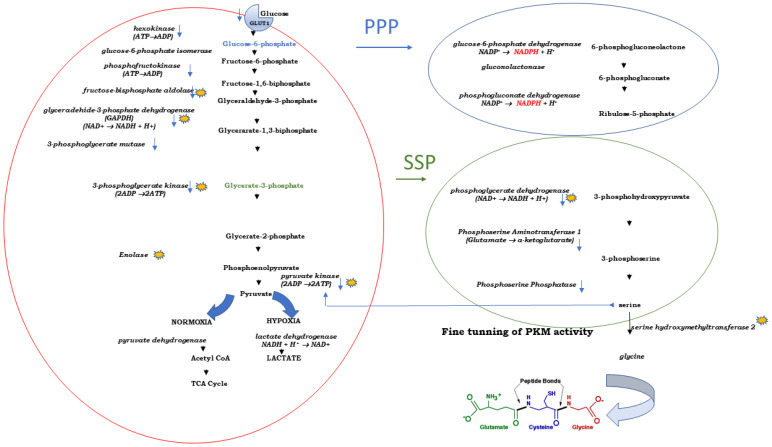
Curcumin modulates the glycolytic metabolic pathway and has the potential to modulate de novo serine synthesis pathway. Blue tiny arrows: decrease of transcript/protein/enzyme activity. Stars: direct binding of curcumin to the target protein.

**Figure 2 pharmaceutics-15-01612-f002:**
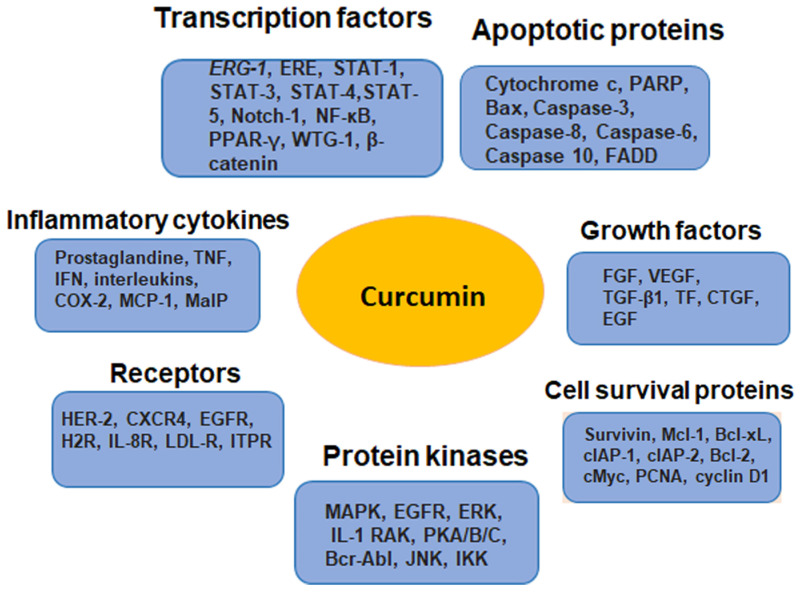
Pleiotropic effects of curcumin relevant for its chemosensitizing activity.

**Table 1 pharmaceutics-15-01612-t001:** The main studies reporting in vivo curcumin effects on experimental tumor models and cancer patients.

Major Findings	Tumor Type	Adm. Route	Hist.	Ref.
Symptomatic relief, ↓ lesion size	Patients with external cancers	topical	−	[5]
Tumor growth inhibition	Head and neck carcinoma (M)	topical	−	[24]
↓ CD31 and HIF-1α expression	Head and neck carcinoma (M)	p.o.	+	[25]
↑ IFN-γ and granzyme expression	Head and neck carcinoma (M)	i.p.	+	[26]
↑ T-cell proliferation, ↓ PD-1 expression	Head and neck carcinoma (M)	i.p.	+	[27]
Humoral response (+immunization)	Melanoma B16-R (M)	i.p.	−	[28]
↓ ATP-synthase activity, ATP/AMP ratio	Melanoma B16-R (M)	i.v.	+	[29]
↓ Foxp3^+^ Tregs in tumor	Melanoma B16-F10 (M)	i.p. GO-Y030 *	−	[30]
↓ Tumor formation	Dalton’s lymphoma (M)	i.p. (liposomes)	−	[4]
Downregulation of DNMT1	Acute myeloid leukemia (M)	i.p.	−	[31]
↓ MMP2, MMP9, vimentin expression	Monocytic leukemia (M)	i.p.	+	[32]
↓ EZH2, H3K4me3, H3K27me3	Myelodysplastic syndrome (M)	-	−	[33]
Apoptosis induction	Colon adenocarcinoma (R)	p.o. (diet)	+	[34]
Inhibition of colon tumorigenesis	Colon adenocarcinoma (R)	p.o. (diet)	−	[35]
Tumor prevention, ↑ CD4^+^ T-cells, B cells	Genetic colon cancer (M)	p.o. (diet)	+	[36]
Inhibition of proteasome activity	Colon cancer HCT-116 (M)	p.o.	+	[37]
↓ PCNA, β-catenin, Axin-2	Colon cancer (M)	?	+	[38]
↑ Oxidative stress, mitochondrial Ca^2+^	Colorectal cancer (M)	i.p.	+	[39]
Inhibition of tumor-induced Immune cell ↓	Ehrlich’s ascites carcinoma (M)	p.o.	+	[40]
No peritoneal bulge + survival	Histiocytic tumor AK-5 (R)	i.p.	−	[41]
ER stress-associated apoptosis	Liposarcoma (M)	i.p.	+	[42]
↓ MDSCs, lL-6, Il-1β, GM-CSF secretion	Hepatocarcinoma HepG2 (M)	p.o.	+	[43]
Inactivation of JAK2/STAT3 pathway	Hepatocellular carcinoma (M)	i.p. GL63 *	−	[44]
Suppression of Wnt/ β-catenin signaling	Gastric carcinoma (M)	p.o.	−	[45]
Inhibition of β-catenin and STAT3	Genetic gastric cancer (M)	p.o. (diet) GO-Y031 *	+	[46]
Activation of ER stress pathway	Adrenocortical carcinoma (M)	i.p.	+	[47]
Induction of cleaved caspase-3 and PARP	Prostate cancer PC-3 (M)	i.p.	+	[48]
↑ Apoptosis, ↓ tumor angiogenesis	Prostate cancer LNCaP (M)	p.o. (diet)	+	[49]
↓ Lung metastasis	Prostate cancer PC-3 (M)	p.o. (diet)	+	[50]
↑ Membrane localization of β-catenin	Prostate cancer C4-2 (M)	intratumoral	+	[51]
NF-κB and p-STAT3 suppression, ↓ Il-8	Ovarian cancer (M)	p.o.	+	[52]
Synergically↑ IFN-β inducedapoptosis	Breast cancer (M)	p.o.	−	[53]
↓ Tumor incidence, DNA adducts	Mammary carcinoma (R)	i.p.	−	[54]
Inhibition of SERCA2, ER stress	Breast cancer (M)	i.p. (RL71) *	+	[55]
Inhibition of STAT3 phosphorylation	Breast cancer (M)	i.p. Curcumin-BTP hybrid *	+	[56]
N-cadherin, MMP2, MMP9 suppressed	Breast cancer (M)	i.p. WZ35 *	+	[57]
↓ HIF1α/mTOR/VEGF cascade	Lewis lung cancer (M)	?	−	[58]
↑ Expression of FOXO3a, p27, p21	Lung cancer A549 (M)	i.p.	+	[59]
Activation of p53-miR-192-5p/215-XIAP	Lung cancer (M)	p.o.	+	[60]
↓ Circ-PRKCA, ITGB1 expression	Lung cancer A549 (M)	p.o.	−	[61]
Induction of ferroptosis via autophagy	Lewis Lung carcinoma (M)	i.p.	+	[62]
Immune response (CD8^+^ T cells), ↓ Il-6	Mesothelioma (R)	i.p.	+	[63]
↑ p38, MAPK and CARP-1	Patient mesothelioma (M)	p.o.	+	[64]
Proteome changes (liver invasion)	Mesothelioma (R)	i.p.	+	[65]
Proteome changes (residual tumors)	Mesothelioma (R)	i.p.	+	[66]
↓ Tumor volume, hemorrhage	Glioblastoma C6 (R)	i.p.	+	[67]
↑ PTEN and P53 expression	Glioblastoma U87 (M)	i.p.	+	[68]
↑ HSP70, ER stress, immune cells	Glioma GL261 (M)	i.p.	+	[69]

* Curcumin analogs. Abbreviations: p.o., per os, i.p., intraperitoneal; i.v., intravenous injection; Hist., histological data included (+), not included (−). Animal models, M (mouse), R (rat).

**Table 2 pharmaceutics-15-01612-t002:** Effect of curcumin on non-tumorigenic cell lines in vitro.

Major Cellular Findings	Cell Line	Formulation	Solvent	Ref.
Cellular viability ↓, proliferation ↓, necrosis ↑, apoptosis ↑	Primary Human Dermal Fibroblast (ATCC^®^PCS-201-012™)	Curcumin (10 μM/24 h) *	DMSO	[93]
Lack of apoptosis	Primary cultures of: rat hepatocytes, lymphocytes and skin fibroblasts, Chinese hamster ovary cells	Curcumin (50 μM/24 h) **	Ethanol	[94]
Reversible inhibition (4–24 h) of cell cycle progression without apoptosis	Normal Human Mammary Epithelial Cell Clonetics^®^	Curcumin (10 mM/48 h) **	Unknown	[95]
Lack of apoptosis	Human lung epithelial (Beas2b) and prostate epithelial (PrEC) cells	Curcumin (30 μM/48 h) **	DMSO	[96]
No change in cellular viability	Normal human gingival fibroblasts (HGF) and normal human oral keratinocytes (OKs)	PLGA nanoparticles loaded with curcumin (80 μM/48 h) *	-	[97]
Lack of apoptosis	Immortalized cell lines: Vero (kidney of a normal adult African green monkey); F111 (rat)	Curcumin (50 μM/24 h) **	Ethanol	[94]
No change in cellular viability	Immortalized human fibroblasts WI-38	Curcumin (50 μM/72 h) **	Ethanol	[98]
Cellular viability ↓	Immortalized embryonic human kidney cell line HEK293	Curcumin (20 μM/72 h) *	DMSO	[96]
Minimal change in cellular viability (more than 90% of viable cells)	Non-tumorigenic mesothelial rat cell line F1-0e	Curcumin (75 μM/4 h and 50 μM/6 h)	DMSO	[63]

* minimal effective concentration and exposure time; ** maximal applied concentration and exposure time.

## Data Availability

Not applicable.

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
