# Peer review of "Curcuminoids as Anticancer Drugs: Pleiotropic Effects, Potential for Metabolic Reprogramming and Prospects for the Future"

_pharmaceutics, 2023, doi:10.3390/pharmaceutics15061612_

Round 1

Reviewer 1 Report

I would like to see a further discussion, one or two paragraphs, of future projections: What might be the eventual human clinical translation of this research? In what time horizon? What will its impact be in an optimal scenario?

Otherwise, extensively reviewed, and the only thing it needs is some fine tuning of the syntax, numerous words are joined in one at different manuscript lines 

One may consider it a boring retelling of all published knowledge on the subject. But it is an extensive subject that may become a reference and gain references. 

Reviewer 2 Report

The manuscript "Curcuminoids as anticancer drugs: pleiotropic effects in vivo, potential for metabolic reprogramming and prospects for the future" shows information on curcumin and its derivatives with antitumor potential. However, authors should address the following points:

· The authors must modify the title since in vitro results are shown in the manuscript.

· The introduction of structures indicating the isolated, semi-synthetic and synthetic curcuminoids to date would be relevant. Indicating in which phase of study it is.

· A separate analysis indicating whether there is a specific chemosensitisation on specific tumour lines would be highly relevant.

· Although the authors collected information on curcuminoids and their effects on tumour lines. Is there information on the effects of this type of compound on non-tumour lines? In addition to those already mentioned in microenvironments or cell-cell interactions.

They must standardise the language, there are words in British English and others in American English.

Reviewer 3 Report

Overall, this work provides a comprehensive overview of the growing body of research on curcuminoids in cancer research, and highlights their potential as chemosensitizing agents and modulators of metabolic reprogramming. However, there are several areas in which the manuscript could be improved:

  1. The introduction could be strengthened by providing more context for why curcuminoids are of interest in cancer research, and by clearly stating the research questions that the review aims to address.
  2. While the authors provide a detailed chronology of discoveries related to curcuminoids, the discussion could be more focused on the most recent and relevant findings, and how they contribute to our understanding of the potential applications of curcuminoids in cancer treatment.
  3. The section on metabolic reprogramming could benefit from a more detailed explanation of what this term means and how it is relevant to cancer research, as well as more specific examples of how curcuminoids modulate metabolic pathways.
  4. The section on chemosensitization could be more informative by providing more details on the mechanisms by which curcuminoids enhance the efficacy of anticancer drugs, and by highlighting any challenges or limitations to this approach.
  5. The conclusion could be more robust by summarizing the key findings and contributions of the review, and by providing a clear roadmap for future research in this area.

Minor comments:

Figure 1 is too complex to understand it needs to be improved and provided in a more simpler manner to improve readability 

Figure 2 needs improvement its difficult understand the message being conveyed via this image.

Overall, while the manuscript provides a useful overview of the current state of knowledge on curcuminoids in cancer research, it could benefit from a more focused and informative discussion of the most recent and relevant findings, as well as more thorough explanations of key concepts and mechanisms.

Round 2

Reviewer 3 Report

The authors have provided rebuttal for all my comments, I have no further comments.

No further comments on english, however I will request for a final proof reading.